# Evaluating the Anti-Osteoporotic Potential of Mediterranean Medicinal Plants: A Review of Current Evidence

**DOI:** 10.3390/ph17101341

**Published:** 2024-10-08

**Authors:** Alhareth Abdulraheem Al-Ajalein, Nurul ‘Izzah Ibrahim, Mh Busra Fauzi, Sabarul Afian Mokhtar, Isa Naina Mohamed, Ahmad Nazrun Shuid, Norazlina Mohamed

**Affiliations:** 1Department of Pharmacology, Faculty of Medicine, Universiti Kebangsaan Malaysia, Jalan Yaacob Latif, Bandar Tun Razak, Cheras, Kuala Lumpur 56000, Malaysia; p137126@siswa.ukm.edu.my (A.A.A.-A.); nurulizzah@ukm.edu.my (N.‘I.I.); isanaina@ppukm.ukm.edu.my (I.N.M.); 2Department of Tissue Engineering and Regenerative Medicine, Faculty of Medicine, Universiti Kebangsaan Malaysia, Cheras, Kuala Lumpur 56000, Malaysia; fauzibusra@ukm.edu.my; 3Advance Bioactive Materials-Cells (Adv-BioMaC) UKM Research Group, Universiti Kebangsaan Malaysia, Bangi 43600, Malaysia; 4Department of Orthopaedics and Traumatology, Faculty of Medicine, Universiti Kebangsaan Malaysia, Kuala Lumpur 56000, Malaysia; drsam@ppukm.ukm.edu.my; 5Department of Pharmacology, Faculty of Medicine, Universiti Teknologi Mara (UITM), Jalan Hospital, Sungai Buloh 47000, Malaysia; anazrun@uitm.edu.my

**Keywords:** mediterranean medicinal plants, bones, osteoporosis, osteoporotic fractures, animal model

## Abstract

**Background:** Bones are biological reservoirs for minerals and cells, offering protection to the other organs and contributing to the structural form of the body. Osteoporosis is a prevalent bone condition that significantly impacts people’s quality of life. Treatments utilizing natural products and medicinal plants have gained important attention in the management of osteoporosis and its associated implications, such as osteoporotic fractures. Even though thousands of plants grow in the Mediterranean region, the use of medicinal plants as an alternative therapy for osteoporosis is still limited. **Methods:** This article provides a comprehensive overview of seven Mediterranean medicinal plants that are used in osteoporosis and osteoporotic fractures in in vitro, in vivo, and clinical trials. The mechanism of action of the medicinal plants and their bioactive compounds against diseases are also briefly discussed. **Results**: The findings clearly indicate the ability of the seven medicinal plants (*Ammi majus*, *Brassica oleracea*, *Ceratonia siliqua* L., *Foeniculum vulgare*, *Glycyrrhiza glabra*, *Salvia officinalis*, and *Silybum marianum*) as anti-osteoporosis agents. Xanthotoxin, polyphenols, liquiritin, formononetin, silymarin, and silibinin/silybin were the main bioactive compounds that contributed to the action against osteoporosis and osteoporotic fractures. **Conclusions:** In this review, the Mediterranean medicinal plants prove their ability as an alternative agent for osteoporosis and osteoporotic fractures instead of conventional synthetic therapies. Thus, this can encourage researchers to delve deeper into this field and develop medicinal-plant-based drugs.

## 1. Introduction

Bone is a multidimensional and hierarchically organized structure that includes a network of cells and their correlated cellular activities [1]. Bones are essential for regulating mineral homeostasis, and they house the bone marrow and maintain the size and shape of the skeleton [2]. Various factors can contribute to the development of bone diseases and their complications. These factors include cancers and related medications, inadequate consumption of nutrients and vitamins, accidents, infections, consumption of certain drugs like glucocorticoids, imbalances in sex steroids and parathyroid hormones, and detrimental lifestyles such as smoking and alcohol consumption, as well as specific diseases like chronic kidney disease and rheumatoid arthritis. Bone fracture refers to the occurrence of a complete or partial cut in the anatomical structure of a bone, resulting in a reduction in the bone’s mechanical strength. This complication is usually related to accidents and osteoporosis [3,4,5,6,7,8,9,10]. This review focuses on osteoporosis, which is noticeable across a large population of people over 65 and postmenopausal women.

Osteoporosis is a significant bone disorder, especially in postmenopausal women and the elderly population. In the USA, osteoporosis is responsible for causing fractures in around 2 million cases, making it the most common cause of fractures, placing above accidental fractures [9]. Osteoporosis is related to decreased bone density and mechanical strength, which makes fractures more possible. The problem is mostly associated with the dysregulation or imbalance between the processes of bone formation and bone resorption [11]. Estrogen deficiency in women and low testosterone in men are among the most important factors associated with decreased osteoblast formation [12]. In cases of estrogen insufficiency or aging, oxidative stress may be elevated, leading to a reduction in antioxidant enzymes such as catalase and glutathione peroxidases [13]. Moreover, oxidative stress induces harm to the mechanisms responsible for osteoblast production. The initial treatment for osteoporosis involves the administration of anabolic agents, such as teriparatide and abaloparatide, as well as anti-resorptive agents, including odanacatib and bisphosphonates. Bisphosphonates are particularly regarded as the most effective option [14]. Nevertheless, the prolonged utilization of these synthetic medications results in adverse effects, including osteonecrosis of the mandible (ONJ), morphea-like plaques, gastrointestinal disturbances, and feelings of nausea [15,16,17]. Therefore, there has been a significant increase in the demand for natural products and herbal medicines [18].

Humans have widely used herbs for therapeutic purposes for decades. The use of medicinal plants by Chinese, Indian, and Southeast Asian Ayurvedic specialists is reflected in their societies and has gained interest from advanced Western medicine [19]. This alternative medicine has effectively treated many diseases, such as cardiovascular disease, viral infections, and hypertension [20,21,22]. This is primarily due to the abundance of natural components, including flavonoids and phenolic compounds, which exhibit various activities such as antioxidant, anti-inflammatory, anti-viral, and anti-tumor properties [23]. In the context of bone health, numerous plant species have exhibited the capacity to improve the rate of bone formation and remodeling and promote bone healing through different studies, including in vivo, in vitro, and clinical studies. The plants mentioned are *Aloe vera*, *Morinda citrifolia*, *Syzygium aromaticum*, and *Alpinia officinarum* [24]. Medicinal plants are abundant in the Middle East and North Africa (MENA) and in southern Europe. All these regions are called the Mediterranean regions. The various climates across this large region make the region a rich source of different medicinal plants [25,26]. Figure 1 shows examples of Mediterranean medicinal plants used to treat different diseases such as tuberculosis, cardiovascular disorders, epilepsy, depression, and bone disorders such as osteoporosis [27].

The economic impact of treating osteoporosis and fractures on healthcare systems is stressful and costly across different countries. The consumption of medicinal plant species native to the Mediterranean regions for managing and treating osteoporosis is seen as an alternative to conventional therapeutic approaches. Utilizing plants from the Mediterranean areas with anti-osteoporosis activity would be advantageous regarding economic, environmental, and lifestyle aspects. The Mediterranean regions are home to around 10,000 plants [28]. However, studies on their potential in bone diseases, especially osteoporosis, are constantly increasing. In this article, we have chosen a few Mediterranean medicinal plants and discussed them in detail to ensure the review remains focused. The objective of this review is to elucidate the significance of seven medicinal plants (*Ammi majus*, *Brassica oleracea*, *Ceratonia siliqua* L., *Foeniculum vulgare*, *Glycyrrhiza glabra*, *Salvia officinalis*, and *Silybum marianum*) found in Mediterranean regions and to examine the literature that elucidates their potential as anti-osteoporosis agents, in vitro, in vivo, and in clinical trials. In producing this article, we used various databases such as Google Scholar, PubMed, and Web of Science to search for original articles to be included. This review also seeks to present an overview of the anatomy of bones and the fracture healing process. In addition, it provides information regarding the commercial therapies that have been used in osteoporosis along with their side effects. The background of the medicinal plants is also provided in this article alongside medicinal plants found in different regions that are related to osteoporosis treatment.

## 2. Methodology

This review was conducted using published articles and studies that investigated the anti-osteoporosis properties of Mediterranean medicinal herbs. An overview of bone anatomy and some information about fractures are included. Fracture healing and its different stages are presented, and the current and commercial therapies related to osteoporosis are also discussed. Finally, we have added the historical background of medicinal herbs and their relation to osteoporosis. Carrubba and Scalenghe. (2012) reported that a total of 73 species have been traditionally utilized in the Mediterranean region for their nutritional, cosmetic, bioactive, and medicinal properties [29]. Using various databases such as Google Scholar, PubMed, and Web of Science, we have selected seven medicinal plants to clarify their anti-osteoporosis properties, as supported by in vitro and in vivo studies for all the plants and two clinical trials for two of them. The following keywords were used to identify these plants: Mediterranean medicinal plants; bones; osteoporosis; osteoporotic fractures; and animal model.

## 3. Bone Anatomy and Osteoporotic Fractures

Bone has a porous bio-ceramic structure. Its organic and inorganic components give it a unique advantage, as it has high strength and resistance to fracture [30]. Two structures comprise bones: compact (cortical) and cancellous (spongy) or trabecular. The cortical region forms the outer part of the bone, constituting about 80% of the total mass. The shaft (diaphysis) is the fundamental location where the cortical bones are prominently shown. The canal systems inside the cortical region encompass connective tissue and blood vessels, commonly called Haversian systems. In addition, cortical bones serve as a shield enclosure for trabecular bones. Two types of membranous structures surround compact bones: the endosteum and the periosteum. These layers consist of bone cells, namely osteoclasts, osteoblasts, and osteocytes. The compact bone exhibits a narrow surface area and a greater density than cancellous (spongy) or trabecular bones. The cancellous structure is a porous-like structure that forms 20% of the total mass and exists mainly in the metaphyseal area. The porosity is attributed to the three-dimensional structure of the interconnected bony meshwork (trabecula) and its interspersion with bone marrow. The design of the trabecular bone enhances its flexibility and capacity to endure mechanical stress. However, cortical bones exhibit more excellent resistance to forces due to their high concentration of tissue and biomaterial components, such as lamellar, organic (collagen), and inorganic (hydroxyapatite) components [29,30,31,32].

In typical situations, bone is a regenerative organ that undergoes a process known as bone remodeling. Typically, this process initiates when osteoclast precursors in the peripheral circulation migrate to the site of weakness or damage. Osteoclast precursors differentiate into multinucleated osteoclasts, a process regulated by various factors, receptors, and molecules (Table 1). Mature osteoclasts secrete actin ring and proteolytic enzymes, in addition to hydrochloric acids, to facilitate the degradation and resorption of bone tissue. Following the resorption process, osteoclasts go through apoptosis and initiate bone formation. This process is attributed to a unique anabolic agent referred to as osteoblasts. Mesenchymal stem cells (MSCs) trigger the production of osteoblast precursors. Binding the transcription factor (Cbfa1) with osteoblast genes, such as type 1 collagen, alkaline phosphatase, and osteocalcin, is crucial in converting osteoblast precursors into mature osteoblasts. In Table 2, more factors that govern and encourage the differentiation of osteoblasts are presented. Osteoblasts secrete the matrix onto the surface of the bone, which then undergoes mineralization. The remaining osteoblasts undergo an upgrade into osteocytes, constituting approximately 90–95% of the cellular composition within the bone matrix [30,32,33,34].

Fractures are a prevalent form of injury that affect the human population [51]. Fractures might be particularly difficult or complex to repair due to structural characteristics, which are characterized by inhomogeneity and hierarchy, as well as the composite nature of bones [52]. Fractures may occur as a result of exposure to severe trauma, such as those resulting from traffic accidents, and these are referred to as high-trauma fractures. Fractures may also arise due to weak trauma, which is attributed to osteoporosis [53].

Osteoporotic fractures, also known as fragility fractures, are a common occurrence among postmenopausal women and elderly people [53,54]. Fractures are regarded as the most common scenario associated with osteoporosis. The process of fracture healing can occur without the production of scars. Researchers have classified fracture healing into two categories: direct and indirect fracture healing, which are determined based on histological changes [55] Primary or direct fracture healing is intramembranous bone production without developing a callus. This implies that osteoblasts and osteoclasts can directly facilitate the healing process. Osteon or Haversian systems can create networks of connective tissue and blood vessels between the fracture site and the gap. This form of healing required the fracture to exhibit complete stability and minimal inter-fragmentary displacement. On the other hand, secondary or indirect fracture healing includes the processes of intramembranous and endochondral bone production. This form of healing is more prevalent than direct fracture healing. It begins when the fracture is relatively stable, and there is movement between the fractured bones, or inter-fragmentary movement [55,56,57].

## 4. Fracture Healing Process

As stated earlier, indirect fracture healing is a more common and complex process than direct fracture healing. Indirect fracture healing is divided into three phases: the inflammatory phase, soft and hard callus formation, and bone remodeling [58] (Figure 2).

### 4.1. Inflammatory Phase

The inflammatory response is widely recognized as a crucial player in initiating the fracture healing cascade [59]. When a fracture occurs and causes damage to the blood vessels in the bone and the surrounding soft tissue, a fracture hematoma is formed. Hematomas serve as scaffolds that create an optimal environment for the differentiation of mesenchymal stem cells (MSCs) and facilitate several biological processes, including vascular development (angiogenesis) and inflammatory activity [60]. During the 24 h following a fracture, many cytokines, including tumor necrosis factor-alpha (TNF-a) and interleukins (IL-18, 1b, 6, and 1a), as well as inflammatory cells such as lymphocytes, macrophages, and neutrophils, together with growth factors, are located in the fracture site [61,62]. All of these factors contribute to the production of osteoblasts by MSCs, the acceleration of apoptosis of hypertrophic chondrocytes, and the stimulation of osteoclast secretion from either macrophages/monocytes or osteoblasts [63]. At this stage, bone regeneration occurs directly via mesenchymal stem cells (MSCs) if the fracture is stabilized. This part of healing is referred to as intramembranous ossification. When a fracture is not fixed, it undergoes a further process known as endochondral ossification [64].

### 4.2. Callus Formation

After an inflammatory response and the formation of a hematoma, the injury site is filled with fibrin-rich granulation tissue and fibrovascular tissue instead of a hematoma [65]. The hypoxic situation at the fracture site and other factors, such as microenvironmental factors and macrophage-derived signals, facilitate the differentiation of MSCs, producing chondrocytes. The collaboration of chondrocytes with fibrous tissue results in a cartilaginous or soft callus. The soft callus acts as a scaffold for the occurrence of endochondral bone and is also considered the initial mechanical support at the fracture site [63]. The soft callus undergoes expansion to form a bridge at the end of the fracture. While the chondrocytes are in the hypertrophic stage, they exhibit an expanded volume and an increase in dry mass. Hypertrophic chondrocytes possess collagen X, which serves as an enhancer in mineralization. Moreover, vascular invasion facilitates the production of calcified callus by the effects of different growth factors, including placental growth factor (PIGF), vascular endothelial growth factor (VEGF), and platelet-derived growth factor (PDGF). This process involves molecular events that finally lead to calcification [65,66,67]. 

### 4.3. Bone Remodeling

At this stage, the crucial aspect is the equilibrium between the activities of osteoblasts and osteoclasts [68]. Osteoclasts initiate the process of reabsorbing hard calluses and replacing them with newly formed compact bones at the core of the callus, as well as lamellar bones around the edge of the callus, which are created by osteoblasts. The remodeling process is crucial for rebuilding the fractured bone to its original form and providing structural support for the utilized mechanical force [69].

## 5. Current and Commercial Therapies in Osteoporosis

Osteoporosis, along with its associated consequences such as osteoporotic fractures, is a significant challenge to an array of humanity, particularly the elderly and postmenopausal women. Synthetic therapies are critical and regarded as the primary approach for treating osteoporosis, maintaining bone mass, and decreasing complications such as fractures [70]. The key factor in the appearance of osteoporosis is the imbalance between bone formation and resorption. Based on this concept, osteoporosis therapies have been developed and are divided into two main types: anti-resorptive agents and anabolic agents. Anti-resorptive drugs function as regulators of osteoclast activity. They have proven to be effective in managing bone diseases such as osteoporosis and are commonly used as the first line of the defense strategy [71]. Bisphosphonates (BPs) and denosumab are common types of anti-resorptive agents. Bisphosphonates inhibit bone resorption in two ways: (1) complexes of BPs with nitrogen that inhibit guanosine triphosphatases (GTPases) by inhibiting mevalonate pathway components to prevent osteoclast survival and (2) non-nitrogen complexes that bind with the mineral stage to stimulate osteoclast apoptosis [72]. Denosumab acts as a monoclonal antibody that binds with RANKL, blocking the RANK/RANKL signaling pathway and inhibiting the resorption activity [73]. On the other hand, anabolic agents specifically target osteoblasts and promote bone formation. Anabolic agents have demonstrated their efficacy according to their superior outcomes compared to the antiresorptive agents; however, the use of anabolic drugs is still limited, usually due to their high costs [74]. Abaloparatide and teriparatide are the only anabolic medications allowed for usage in the United States; however, romosozumab is under investigation [75]. Table 3 summarizes different anabolic and anti-resorptive agents and their targets.

Although synthetic medications are effective, their long-term usage can lead to significant side effects. Teriparatide and romosozumab are required to be taken for a prolonged time, which can lead to an increased risk of osteosarcoma and cardiovascular disorders. Abaloparatide is related to a substantial expense, in addition to the adverse effects found in other anabolic agents [76]. Gastrointestinal events, typical femur fractures, osteonecrosis of the jaw, and esophageal cancer are listed as the main adverse effects of bisphosphonates and denosumab after long-term usage [77]. Thus, there has been a notable increase in the demand for natural products. These products have the potential to be vital sources of therapeutic agents. Medicinal plants have been suggested as a major source of natural products throughout the decades [78].

**Table 3 pharmaceuticals-17-01341-t003:** Different types of osteoporosis therapies and their targets.

Drugs (Anabolic Agents and Anti-Resorptive Agents)	Targets	References
Bisphosphonates (anti-resorptive agents): Alendronate and risedronate (short-term doses). Ibandronate and risedronate (long-term doses).	Bisphosphonate and nitrogen complex that inhibits guanosine triphosphatases (GTPases) by inhibiting mevalonate pathway components to prevent osteoclast survival. Complex of bisphosphonate without nitrogen that binds with the mineral stage to stimulate osteoclast apoptosis.	[79]
Denosumab (anti-resorptive effects)	Inhibits bone resorption by binding to the receptor activator of nuclear factor-κβ ligand to reduce activation of osteoclasts and bone resorption (antibody against RANKL).	[73]
Teriparatide and abaloparatide (anabolic agent)	Increasing bone formation rather than decreasing resorption.	[75]
Menopausal hormone therapy (MHT) or estrogen therapy (anabolic and anti-resorptive agent)	Stimulating osteoblasts and inhibiting osteoclasts.	[80]
Odanacatib (anti-resorptive agent)	Inhibit Cathepsin K (an osteoclastic enzyme that degrades collagens).	[15]
Saracatinib (anti-resorptive agent)	Inhibit c-src kinase (an enzyme involved in osteoclast activation).	[10]
BHQ 880 (anabolic activity)	Antibody against Dickkopf-1 (inhibitor of the Wnt/β-catenin pathway).	[10]
AMG 785 (anabolic activity)	Antibody against Sclerostin (inhibitor of the Wnt/β-catenin pathway).	[10]
Selective estrogen receptor modulators (SERMs): raloxifene (anti-resorptive agent)	Osteoclast inhibition.	[81]
Parathyroid hormone PTH (anabolic agent)	PTH binds with osteoprogenitor cells, PTHrP, and Indian Hedgehog to chondrocytes formation. PTH binds with Wnt signaling to stimulate osteoblast formation	[79]
Sclerostin blocking (anabolic agent)	Osteoblast formation and increased bone strength.	[82]

## 6. A Historical Overview of Medicinal Plants and Osteoporosis

From ancient civilizations to our days, medicinal plants have been and remain an essential resource for living for humans and organisms. Humans have used herbs for decades to help them in their lives, for example, for food, energy, clothing, and health. Plants were the sole source for all of them [83]. Eighty percent of medications have been fabricated using herbal medicine, in which medicinal plants are considered one of the primary sources of drug discovery. Examples of plant medications are aspirin extracted from willow bark, pseudoephedrine extracted from *Ephedra sinica* Staph, and nabilone extracted from *Cannabis sativa* L. Earlier humans used plants without a precise method for their safe use, just based on sensory investigations (smell, sight, and touch). Now, specialists rely on advanced techniques to use these plants based on analysis methods such as HPLC, LC-MS, FTIR, and UV–Vis [83,84] These methods have successfully determined the bioactive compounds that can treat different diseases [85,86]. In general, plants or any biological system are divided into two main metabolites (primary and secondary). Primary metabolites (PMs) are responsible for development and growth, which are chemical compounds such as fatty acids, carbohydrates, lipids, and proteins. Secondary metabolites (SMs) are more specific and are responsible for the survival of the plants through involvement with the surrounding area. Bioactive compounds are some of the SMs that interact with biological systems. Phenolic compounds (hydroquinone, coumarins, resorcinal, etc.), alkaloids (nicotine, morphine, quinine), and terpenoids are the main bioactive compounds. These compounds act with environments as anti-inflammatory, antioxidant, anticancer, and immunomodulatory agents and have cardiovascular protective effects [86,87,88,89] Bioactive compounds are embedded with residue (cellular marc); thus, several extraction methods have been developed. Extraction methods are generally divided into two main categories: conventional and non-conventional. Based on the main objective of a project, the best way to extract the targeted bioactive compound can be determined. For example, conventional methods (soxhlet extraction and maceration) may be helpful for the activity of the bioactive compounds, but they are also related to environmental issues, solvents, and time consumption. On the other hand, non-conventional methods [microwave-assisted extraction (MAE), ultrasound-assisted extraction (UAE), and supercritical fluid extraction (SFE)] may influence the activity by degradation due to radiation or temperature effects [90]. However, non-conventional methods are commonly used now due to them requiring less solvents, reducing time and cost consumption, and being friendly to the environment [91,92]. Inflammation, diabetes, cardiovascular diseases, and cancers are considered chronic medical disorders, and medicinal plants have been successfully used to treat them [93].

Osteoporosis is a silent disease. Its first discovery was 250 years ago, but physicians and researchers began studies and clinical trials 70 years ago. Paleopathology studies have shown that archaeological samples gave similar screening results, such as the cortical thickness in the current population for osteoporosis [94]. Chinese medicine (CM) proactively used herbal medicine to treat osteoporosis. In 1624, a Chinese formula consisting of eight medicinal plants (Lujiaojia, Tusizi, Gouqizi, Guibanjiao, Niuxi, Shudi, Shanyurou, and Shanyao) was developed, which was called the Zuogui pill (ZGP). Previous studies showed that the ZGP inhibited osteoporosis induced by certain drugs like dexamethasone and glucocorticoid by enhancing or suppressing proteins or genes [95,96,97] Table 4 presents a number of medicinal plants from different regions of the world with their anti-osteoporotic properties.

**Table 4 pharmaceuticals-17-01341-t004:** Examples of the anti-osteoporotic effects of medicinal plants throughout different countries.

Scientific Name	Family	Place of Origin	Active Component	Outcomes	References
*Acanthopanax senticosus*	*Araliaceae*	Russia and East Asia	Syringin and eleutheroside E	Increase bone mineral density (BMD). Decrease the production of osteoclasts by inhibiting the NF-kB and MAPK signaling pathways.	[98]
*Erythrina variegate*	*Fabaceae*	India and Southeast Asia	N. D	Increase mechanical properties and decrease bone loss. Histomorphometry analysis showed increases in the area and thickness of the trabecular region.	[99]
*Actaea racemosa*	*Ranunculaceae*	Eastern North America	Isopropanol and rhizomes	Inhibit osteoclastogenesis by inhibiting the NF-κB signaling pathway. Inhibit the production of TNF-a by suppressing lipopolysaccharide.	[100]
*Asparagus racemosus*	*Asparagaceae*	India	Carbohydrates, flavonoids, steroids, organic acids, and saponin glycosides.	Improvements in biomechanical, biochemical, and histological properties in experiments using ovariectomized (OVX) rats.	[101]
*Epimedium*	*Berberidaceae*	Eastern Asia	Flavonoids, lignins, diadzein, icariin, and genistein.	Improvements in bone mineral density (BMD), pain, and impact rate. Improvements in biomarkers, especially alkaline phosphatase (ALP).	[102]
*Allium cepa*	*Amaryllidaceae*	Iran Pakistan	Flavonoids (quercetin), and phenolic acids	Prevents bone resorption. Inhibits osteoclastogenesis. Maintains calcium in bone. Increases secretion of IL-3 and IL-4. Decreases secretion of IL-6 and TNF-α.	[103]
*Cronus mas* L.	*Cornaceae*	Iran	Flavonoids (quercetin), and kaempferol.	Enhances osteoblastic bone formation-related genes such as RUNX2 and ALP. Suppresses bone-resorption-related genes such as Ctsk, Acp5, and Nfatc1.	[104]
*Fructus Malvae Verticillatae*	*Malvaceae*	Eastern Asia.	Polysaccharides and flavonoids.	Anti-resorptive agent (inhibits osteoclasts by suppressing RANKL). Inhibits all markers that relate to the proliferation and differentiation of osteoclast.	[105]
*Cissus quadrangularis*	*Vitaceae*	India	Vitamin C, triterpenes, and flavonoids (quercetin).	Improves fracture healing by enhancing mineral absorption and metabolism.	[106]

## 7. Mediterranean Medicinal Plants with Anti-Osteoporosis Effects

Medicinal plants from various regions have demonstrated their efficacy as anti-osteoporosis agents. Notably, Mediterranean medicinal plants have shown promising potential in this area. Previous studies have revealed that these plants act as regulators of various signaling pathways, increase bone mineral density (BMD), promote osteoblastogenesis, inhibit osteoclastogenesis, and improve biomechanical, biochemical, and histological properties in experiments conducted on animal models. This suggests that these plants could be a valuable resource in the development of new anti-osteoporosis treatments [99,101,104,106,107,108,109,110,111]. However, few studies have been published on the anti-osteoporosis properties of Mediterranean medicinal plants. These plants have been studied in vitro and in vivo to understand their anti-osteoporosis activity. Animal models provide significant insights into bone studies and are frequently considered more appropriate than in vitro experiments. However, researchers still seek in vitro investigations due to the limitations of in vivo studies, such as the variety of animal responses, time consumption, and expenses.

This review presents a comprehensive analysis of seven Mediterranean medicinal plants (*Ammi majus*, *Brassica oleracea*, *Ceratonia siliqua* L., *Foeniculum vulgare*, *Glycyrrhiza glabra*, *Salvia officinalis*, and *Silybum marianum*) from various plant families, together with their in vitro, in vivo, and clinical trials that have been conducted. The cellular, molecular, and structural findings of these seven Mediterranean medicinal plants on bone-related parameters are discussed and summarized in Table 5, providing insights into their effects on bone health. Figure 3 elucidates the molecular mechanisms of the Mediterranean medicinal plants that contribute to their efficacy against osteoporosis.

### 7.1. Ammi majus

This plant belongs to the *Apiaceae* family and is native to the Mediterranean and Arabian Peninsula. It is an important source of coumarins, flavonoids, proteins, and essential oils. *Ammi majus* possesses a lot of pharmacological activities such as antibacterial, anti-inflammatory, and cytotoxic activities. It has been observed to have high efficacy against skin disorders, especially vitiligo [126] Extract from seeds of *Ammi majus*, namely Xanthotoxin (XAT), showed anti-osteoclastogenesis activity in vitro. The study showed that the Xanthotoxin extract inhibited RANKL-induced Ca^2+^ vibration and reactive oxygen species (ROS) production. Bone-marrow-derived macrophages (BMMs) treated with RANKL were used to explore osteoclast differentiation, formation, fusion, and functional activity. The effect of the extract on osteoclastogenesis, particularly on proliferation, was investigated using TRAP staining. The findings showed a reduction in osteoclast count and TRAP function when XAT was administered at concentrations of 0.01 and 1 μM. In addition, a decrease in osteoclast markers, DC-STMAP, and Ctsk were observed. The study also showed that the formation of osteoclasts was suppressed, and the number of nuclei decreased, which led to inhibition in the fusion of osteoclasts. In addition, the implantation of BMM on osteoassay surfaces and bone slices suppressed osteoclast bone resorption activity through XAT treatment. These results correlate with decreased ROS in RANKL-treated BMMs, in which ROS levels were significantly suppressed using XAT. In addition, the frequency of Ca^2+^ oscillations was inhibited by suppressing the calcium signaling (CaMKK/PYK2) using XAT extract [112].

Animal studies using an ovariectomized mice model also showed good results regarding the anti-osteoclastogenesis activity of XAT extract. The results of the structural analysis indicated a reduction in trabecular separation (Tb. Sp), along with an elevation in trabecular bone volume (BV/TV), trabecular number (Tb. N), and bone mineral density (BMD). Cellular analysis showed a decrease in osteoclast surfaces using TRAP staining. Furthermore, the dynamic study confirmed a rise in the bone formation rate [112]. These findings suggest that *Ammi majus*, particularly its XAT extract, is a highly effective treatment for osteoporosis due to its demonstrated ability to decrease bone loss.

Using an ovariectomized mice model, another extract of *Ammi majus*, called methoxsalen (MTS) revealed its ability against osteoporosis. The findings showed improvements in osteoblast-related markers such as osteocalcin, estradiol, alkaline phosphatase (ALP), Runx-2, and osterix. Also, the bone mineral density (BMD) significantly improved. On the other hand, osteoclast numbers and formation decreased according to the TRAP staining test. Also, the Il6 and Nfκb genes related to the inflammatory response were suppressed. All these findings demonstrated the ability of *Ammi majus* to enhance bone formation and inhibit oxidative stress, as well as its inflammatory response [113].

### 7.2. Brassica oleracea

Cabbage (*Brassica oleracea* L. *var*) originated in the Mediterranean region and then became widely distributed across various countries worldwide [127]. Cabbage is considered one type of the *Brassica* family, and it includes Brussel sprouts, broccoli, garden cress, collard greens, mustard, etc. [128]. Cabbage’s health advantages and bioactivities have made it the focus of many epidemiological studies. A variety of bioactivities, including anti-inflammatory, antioxidant, and anticancer properties, might be attributed to the presence of several bioactive compounds such as vitamins, minerals, flavonoids, phenols, glucosinolates, carotenoids, and anthocyanins [127,128,129]. In an osteoporosis study, Kang et al. [114] observed that *Brassica oleracea* (BO) extract proved effective in inhibiting osteoclastogenesis when combined with *Panax ginseng* (PG). In an in vitro study, mixtures of BO and PG at various concentrations (50, 100, 200 μg/mL) were used on RANKL-treated RAW 264.7 cells. The combination proved its strong ability against osteoclastogenesis at 200 μg/mL.

Furthermore, using an ovariectomized mice model (OVX), the combination reduced the number of osteoclasts. The osteoclasts were identified as multinucleated cells by TRAP and H&E staining. The molecular impact of *Brassica oleracea* is not widely researched. Multinucleated osteoclasts are directly linked to the activation of DC-STAMP [130] Inhibition of this mature form of osteoclast is associated with the downregulation of DC-STAMP. Bone mineral density (BMD) and bone weight showed an increase in the group treated with BO and PG [114]. During the postmenopausal stage, an estrogen imbalance leads to variations in biochemical parameters and alters metabolism, such as glucose levels. In this study, BO did not positively affect the increases in glucose. PG directly reduced glucose levels, which was reflected in other biochemical parameters in the blood, such as alanine transaminase (ALT) and aspartate aminotransferase (AST) [131]. This reflects the possibility of preventing triglycerides, fatty acids, and hyperinsulinemia.

### 7.3. Ceratonia siliqua L.

*Ceratonia siliqua* L., commonly known as carob, is indigenous to the Mediterranean region, especially in the Middle East and North Africa. This plant is classified within the *Leguminosae* or *Fabaceae* family. Carob seeds contain many bioactive compounds, such as proteins, polyphenols, flavonoids, tannins, and carbohydrates. The main benefits of carob include antioxidant and anti-inflammatory abilities and additional properties such as anti-diabetic and anti-cancer effects [115]. According to Sharaf et al. [116], carobs have shown superior results as an anti-osteoporotic agent in bone health. Their histopathological results demonstrated an enhancement in cortical bone thickness (shaft) and the trabecular area in the treated group. On the other hand, the Haversian canal area was reduced. This indicates a positive remodeling balance, resulting in a decrease in the depth of resorption and/or an increase in the width of the wall [132].Furthermore, it was observed that the treated group exhibited a higher number of osteocytes. This indicates that the osteoblasts completed the process of forming new bone and converted into osteocytes [133]. The histochemistry results showed that carob-treated OVX rats had a moderate periodic acid Schiff (PAS)-positive reaction in trabeculae and shaft reversal lines (cement lines) [117]. The positive PAS reaction indicates the production of a pigment that is a critical indicator for the presence of type III collagen, which is essential for the repair process. Type III collagen undergoes conversion into type I collagen, which provides increased stability for newly formed bones and mechanical strength [134]. Another study used an ovariectomized rat model to demonstrate carob’s ability to enhance bone mineral density (BMD), particularly in the proximal tibia [117]. The promising histomorphometry results in this study can be attributed by the antioxidant activity of carob, as it is rich in polyphenols, especially flavonoids. Flavonoids are an important enhancer for upregulating estrogen receptor β, leading to increases in osteoblast-related markers such as RUN-X2 and inhibition of the tumor necrosis factor (TNF-α) to suppress oxidative stress.

### 7.4. Foeniculum vulgare

*Foeniculum vulgare*, often known as *fennel*, is a member of the *Apiaceae* family, and it appeared in Mediterranean Europe before spreading worldwide. Screening studies have revealed that *fennel* is a rich source of bioactive compounds, including proteins, phenolics, flavonoids, and volatile contents. These bioactive components contribute to fennel being an antioxidant, anti-inflammatory, and antimicrobial agent [135]. Previous in vitro *and* in vivo studies have shown that the plant exhibits significant potential as an anti-osteoporosis agent. An in vitro study using RANKL-induced osteoclast formation in bone marrow-derived macrophages (BMMs) demonstrated that fennel seeds at concentrations of 0.5, 1, and 2 μg/mL suppressed OC formation. The extract influenced genes associated with osteoclasts and served as the mechanism for their viability. Thus, the extract’s impact was specifically on viability rather than toxicity; therefore, it was deemed toxic at a 200 μg/mL concentration. Thus, OC-related genes (DC-STAMP, NFATc1, TRAP, and c-Src) were inhibited at 2 μg/mL. The study demonstrated that fennel effectively starts suppressing OC differentiation at an early stage. In terms of the OC resorption activity, the number of OCs on bone slices showed no changes in the fennel-treated group. It was noted that the impact on the cells depended on the concentration [117,118]. In vivo investigations were also conducted and focused on parameters including bone loss, blood markers, and mechanical strength. Micro-computed tomography (μ-CT) analysis revealed significant improvements in bone mineral density (BMD), bone mineral content (BMC), cortical bone mineral density (Cr. BMD), and tissue mineral density (TMD) in the treated group (30 or 100 mg/kg/day). Regarding mechanical strength, the experimental groups supplied with fennel seeds (100 mg/kg) exhibited superior strength and maximal load. These findings demonstrate the capacity of fennel to prevent a decline in flexural strength. Both dosages of fennel resulted in a minor reduction in osteoclast resorption activity (C-telopeptide of type I collagen (CTX)). In the blood serum markers test, both dosages appeared to cause a small decrease in osteoclast resorption activity (C-telopeptide of type I collagen (CTX)) [118].

*Fennel* has also shown a superior effect in histological findings as well. In an in vivo study conducted by Sharaf et al. [116], a significant increase in cortical bone thickness was observed in the fennel-treated OVX rat group. This finding is significant, as it highlights fennel’s potential to counteract bone loss typically induced by estrogen deficiency, which is modeled by OVX in rats and mimics postmenopausal osteoporosis in humans. Osteocyte counts showed better results in the group treated with fennel. The trabecular area values exhibited a slight elevation in the group administered with fennel. The periodic acid-Schiff (PAS) staining method showed an excellent positive reaction in the group treated with fennel [116].

The osteoclast-inhibitory effect of fennel extract was at the first stages of OC differentiation (OC precursors). Mature OCs were not directly affected by fennel according to the OC functional test. It was assumed that the effect resulted from the apoptosis of OCs. The fennel extract’s antioxidant, anti-inflammatory, and estrogenic activities demonstrated its ability as a potential therapy for osteoporosis. The binding to the estrogen receptor is responsible for regulating osteoclasts and osteoblasts by the production of specific proteins. Also, the antioxidant activity of fennel seeds was attributed to the suppression of lipid peroxidation [116,133].

While in vitro and in vivo experiments have shown favorable results for *Foeniculum vulgare*, the clinical trial conducted by Ghazanfarpour et al. [136] revealed negative results. The trial possessed a time frame of 12 weeks and involved the participation of postmenopausal women. The results showed negative impacts on bone mineral density (BMD) and bone mineral content (BMC). It is crucial to highlight that the favorable outcomes of the clinical trials, which used natural products for treating osteoporosis, were observed over a longer period, ranging from 6 months to 2 years [136].

### 7.5. Licorice

*Glycyrrhiza glabra*, or *licorice*, belongs to the *Fabaceae* family. This plant has spread throughout Mediterranean countries due to its diverse applications in the pharmaceutical industry, cosmetics, food production, and cigarettes. *G. glabra* is an essential source of pectin, proteins, minerals, gums, starch, monosaccharides, and sterols. Phytochemical investigations have shown that the plant contains liquiritin, isoliquiritin, glycyrrhizin, glycyrrhizinic acid, glabridin, formononetin, and isoliquiritigenin. These compounds are related to various activities such as anti-tyrosinase, antioxidant, anti-diabetic, anti-inflammatory, and anticancer activities, neural protection, and liver protection [137]. Previous studies have shown the effect of *G. glabra* on osteoporosis through in vitro and in vivo approaches. A study by Hong et al. [119] focused on decreasing NFATc1 action in vitro by reducing ROS levels, suppressing osteoclast-related genes, and inhibiting NF-κB induced by RANKL. A *G glabra*-derivative compound, liquiritin, was used on BMMs to determine the osteoclast differentiation and formation measured under the TRAP effect. The liquiritin caused the suppression of OC differentiation from day 3 to 6. The liquiritin also showed an inhibitory effect on mature osteoclast and OC formation. According to an MTS assay, the suppressing effect of the extract was not a cytotoxic effect. An osteoclast resorption activity test showed that liquiritin can inhibit the function of osteoclasts, especially at 0.1 Mm. Intracellular reactive oxygen species (ROS), particularly in osteoclasts, enhance their activity and production by increasing the receptor activator of nuclear factor kappa-B ligand (RANKL). NADPH oxidase 1 (NOX 1) is recognized as the central mediator of ROS synthesis. The study’s results showed an excellent inhibitory effect of the liquiritin (0.01, 0.05, and 0.1 mM) on NADPH oxidase 1 (NOX1) [119]. Furthermore, the liquiritin stimulated enzymes related to antioxidant activity, such as glutathione disulfide reductase (GSR), heme oxygenase-1 (HO-1), and catalase. The activation of the signaling pathway associated with NFATc1 is attributed to the Ca^2+^ oscillations induced by RANKL. The concentration of 0.1 mM liquiritin was found to be remarkably effective in reducing Ca^2+^ oscillations. This significant reduction in calcium oscillations highlights liquiritin’s potential as a modulator of intracellular calcium signaling, which plays a crucial role in various cellular processes. The *G. glabra* extract also showed a significant blocking effect for the osteoclast-related genes, especially at 0.1 mM. The NF-κB and MAPK signaling pathways are considered important pathways in regulating bone homeostasis. A Western blot analysis conducted using BMMs cells and a luciferase assay performed on RAW264.7 cells showed that the extract (0.1 mM) effectively inhibited the expression of IκB-α protein, which is associated with the activation of NF-κB. Also, it demonstrated the inhibition of the JNK and p38 proteins, which contribute to MAPK phosphorylation. In terms of the NFATc1 pathway, a luciferase assay was utilized using RAW264.7 cells with RANKL, and the results showed the inhibition of NFATc1 activity stimulated by RANKL based on the dose amount. Also, the Western blot method was conducted using BMM cells to measure the level of downstream protein factors such as c-fos, V-ATPase-d2, and integral α. The results showed a remarkable decrease in these proteins.

In vivo studies have demonstrated promising results in histomorphometry, micro-CT, and blood serum markers. Based on the histological analysis, it was observed that the BV/TV of the OVX group treated with liquiritin exhibited an increase. Additionally, the cellular analysis indicated a decrease in N. Oc/BS and Oc. S/BS in the liquiritin-treated group. The micro-CT test confirmed an increase in BV/TV and a decrease in TbSp. The blood serum level analysis showed the inhibition of CTX-1 and TRAcP in the treated OVX group [119]. The in vivo results revealed that the extract was able to effectively suppress the formation of osteoclasts and bone resorption.

Galanis et al. [120] studied the mechanical strength and bone mineral density (BMD) of ovariectomized rats using another extract named glycyrrhiza. A dual-energy X-ray absorptiometry (DEXA) scan was utilized to measure the bone mineral density (BMD) of the total tibia and proximal tibia. The absolute values of the total tibia at three months indicated an improvement in the treated group. At six months, the treated group had a small decline in BMD but remained higher than others. The BMD at the proximal tibia site showed the ability of the glycyrrhiza as an antiosteoporosis agent. According to the results, the treated OVX group showed values similar to those of the positive control group at three months, while at six months, it showed values close to those of the baseline and positive control groups. The percentage changes between the baseline and the positive control and treated groups for the whole and proximal tibia were insignificant. On the other hand, the changes in the values of the BMD percentage, especially from baseline to 6 months, between the treated group and the OVX control group were significant. A mechanical strength test was applied at the shaft of the femoral bone. The results did not indicate any significance in the treated group even though the force used for the fracture was slightly low. It is expected that this result appeared due to the femoral shaft or diaphysis structure, which contains cortical bone; thus, the treatment’s efficacy at this site was low [120].

Another study used glycyrrhizinic acid and glabridin as target components of *G. glabra*. However, in this in vivo study, these two components did not show any significant results on the mechanical strength and chemical contents of the bone using an OVX rat model [138]. Kaczmarczyk-Sedlak et al. [121] used a different component of *G. glabra*, namely, formononetin. They observed slight improvements in chemical contents (noticeable decrease in water content and slight increase in mineral content) and mechanical strength (increases in fracture load and maximum load versus decreases in displacement fracture load and maximum load) [121].

### 7.6. Salvia officinalis

This plant belongs to the *Lamiaceae* family and is native to the Mediterranean region and the Middle East. *Salvia officinalis* has spread to the North Atlantic Ocean due to its various uses in food industries, where it has a good flavor and medical uses. Over several years, plants have been used to treat several medical conditions, including inflammation, dyspepsia, ulcers, hyperglycemia, gout, and other disorders. The biological activities of *S. officinalis* that were explored through previous studies, such as anti-inflammatory, antimicrobial, antioxidant, anti-mutation, and other activities, are related to treating different medicinal conditions, as mentioned earlier. *S. officinalis* contains a lot of bioactive components, such as polyphenols, carbohydrates, terpenoids, steroids, flavonoids, and alkaloids. The presence of these components is dependent upon the specific type of extract, namely, methanolic, aqueous, alcoholic, and essential oil extracts. However, most of these constituents have been detected in essential oils [139]. The anti-dopaminergic activity of *Salvia officinalis* can decrease menopausal complications such as estrogen deficiency [140]. According to Kayalar et al. [141], *S. officinalis* has shown activity against bone metabolic disorders and distressing bone fractures. The plant can raise bone mineral density and decrease bone resorption activity [141]. In a study by Abd El-Motelp et al. [122], it was observed that a group treated with *Salvia officinalis* exhibited significant improvements in all evaluated parameters. The blood serum calcium and phosphorus levels exhibited notable enhancements. Vitamin D and estrogen (E2) exhibited similar findings. On the other hand, parathyroid hormone (PTH) showed a reduction in the treated group. Moreover, *Salvia officinalis* demonstrated a reduction in all markers of bone resorption (RANK, BALP, PINP, and CTX-1). A histological study on the femur of rats showed that the treated group had improvements in the trabeculae shape with the existence of osteoid lines (reversal and resting), indicating the presence of new bone formation beside the old bone. On the other hand, the OVX control group showed a deterioration in the bone matrix and mineralization. In the investigation into the immune response, it was observed that the treated group exhibited a modest production of TNF-α cytokine, which is known to stimulate osteoclasts and inhibit osteoblasts. In contrast, the OVX control group had significantly elevated TNF-α levels [122].

The only clinical trial to test *Salvia officinalis* as an anti-osteoporosis agent was performed by Zeidabadi et al. [142]. Postmenopausal women were selected based on specific criteria. In this trial, calcium, phosphate, and vitamin D were tested as osteogenic factors. The findings revealed that *Salvia officinalis* effectively raised the calcium and phosphate levels in blood [142]. The findings highlighted the phytoestrogenic properties of *Salvia officinalis*, which successfully treated the reduction in estrogen levels and increased the levels of calcium and phosphate.

### 7.7. Silybum marianum

*Silybum marianum* (SM), or milk thistle, spreads in different regions, especially in the Mediterranean. It belongs to the *Leucanthemum* family, and the backbone of this plant is the flavonolignan compound silymarin. The bioactive compounds of silymarin include silychristin, silydianin, isosilybin, dihydrosilybin, and silybin. SM has been widely used in food and for its pharmacological activities. SM is a potentially helpful agent as an antidiabetic, anti-cancer, immunomodulatory, antimicrobial, hepatoprotective, skin protective, antioxidant, anti-inflammatory agent [143]. In general, the regulation of inflammation and oxidation is mainly governed by SM through many signaling pathways [106,140]. In osteoporosis studies, a methanolic extract of SM, which contains primarily silybin or silibinin, caused enhanced osteoblast markers such as ALP and the inhibition of RANKL activity. These results led to an increase in bone formation activity [107]. In a study by Tao et al. [123], the researchers used hydroxyapatite combined with silymarin in an ovariectomized rat model. The reason for using silymarin was the insufficient activity of hydroxyapatite as an antiosteoporosis agent. The micro-CT investigation showed that the implants coated with silymarin and hydroxyapatite improved the microarchitecture parameters (BV/TV), (Tb. Th), (Tb. N), and (TbSp), which enhanced the bone formation surrounding the implants. During the histomorphometry evaluation, it was demonstrated that the treated group exhibited improvements in all parameters. The bone–implant contact (BIC) and bone area ratio (BAR) results demonstrated that the implants effectively covered newly formed bone, particularly in the treatment group. In terms of the mechanical analysis, the maximum push-out strength increased in the treated group. The findings of this analysis demonstrate the efficacy of silymarin as a viable option for implementation in conjunction with hydroxyapatite titanium implants. The PCR test outcomes indicated higher levels of gene expression related to bone formation (OPG, Runx2, and OCN) in the treatment group [123].

Another compound of silymarin, namely silibinin or silybin, was also observed to have excellent activity against osteoporotic fractures. The extract first underwent in vitro testing using an MC3T3-E1 cell line. The activity of the cells and proteins associated with the HIF-1α/VEGF and Notch signaling pathways was determined using high and low concentrations of silybin. In addition to Western blot analysis, this study also employed two types of staining techniques, Alizarin Red and alkaline phosphatase staining. The results demonstrated a high osteogenic activity of the cells at both high and low concentrations of extract. Additionally, the cells exhibited mineralization. For both the low and high therapy dosages, the Western blot analysis revealed the presence of many proteins that augmented the HIF-1α/VEGF and Notch signaling pathways. These proteins included RUN-X2, OC, Notch1, and ALP. In brief, Notch receptors (1–4) and their legends play a crucial role in bone dynamics, and their efficacy is based on the various differentiation stages. Each receptor has a specific task regarding the interconnection and complementarity of work between receptors. Notch-1 is important for osteoblast precursor proliferation, but the developing process stops at the end of the differentiation (the cell is still immature), leading to mineralization impairment. Also, it affects OPG/RANKL, indirectly reducing the osteoclast. Notch-2 enhances bone resorption by increasing osteoclasts; simultaneously, there are no changes in or influences on the osteoblast number or activity. Notch-3 and -4 promote the upregulation of markers that are related to osteoblasts [144]. Meanwhile, the new blood vessel formation (angiogenesis) at the fracture site starts when the vascular endothelial factor (VEGF) gene upregulates hypoxia-inducible factor-1α (HIF-1α) during a state of hypoxia. It is crucial to acknowledge that the signaling pathways intersect with each other. HIF-1α stimulates the activation of the Notch ligand, specifically, delta-like ligand 4 (Dll4), hence serving as an enhancer of Notch signaling. This complicated mechanism promotes angiogenesis and osteogenesis [145,146]. In vivo, silybin was administered in an ovariectomized rat model. Using micro-CT investigation, all the microarchitecture parameters showed a significant quantity of bone tissues in the treated group. The histopathological analysis showed new bone formation at the defect area in the treated group and a large amount of calcein dye using a fluorescent technique at the defect area. Western blot analysis showed significant VEGF and HIF-1α protein enhancement on the target area in the silybin-treated groups. The PCR analysis revealed the upregulation of the gene expression, especially those related to Notch signaling, such as HEY1 and Notch1, in the silybin-treated groups. The results demonstrated that silybin or silibinin effectively stimulates the activation of the VEGFA/HIF-1α and Notch signaling pathways, leading to an increase in angiogenesis and bone formation [124].

In a diabetic rat model, silibinin showed enhancements in different bone analyses via oral administration. Rats were divided into control, diabetic rats without treatment, and diabetic rats with treatment (50 and 100 mg/kg/day). Bone histomorphometry examinations, including microarchitecture, morphology, and histology, showed that the group treated with 100 mg/kg/day exhibited significant improvements in comparison to both the group treated with 50 mg/kg/day and the non-treated group. The micro-CT analysis included microarchitecture parameters (TbSp, TbTH, TbN, and BV/TV), whereby a significant reduction in TbN and BV/TV levels was observed in both the non-treated group and the group treated with 50 mg/kg/day, whereas the TbSp levels showed a decrease in the group treated with 100 mg/kg/day. The femoral bone exhibited significant increases in height, length, and width within the group treated with 100 mg/kg/day compared to the group treated with 50 mg/kg/day and non-treated group. Eosin and hematoxylin staining showed substantial histological changes in the metaphysis region of the femur. The group treated with 100 mg/kg/day exhibited a higher cancellous (trabecular) bone prevalence than the other groups. The serum biochemical analysis revealed that the levels of the osteoblast development marker alkaline phosphatase (ALP) increased in the groups that received the 100 mg/kg/day treatments [125]. This significant increase in ALP levels indicates enhanced osteoblast activity and differentiation, demonstrating its potential as an effective therapeutic agent for bone health.

## 8. Conclusions

Although advanced therapeutic agents are considered the first line of defense against osteoporosis, their adverse effects are regarded as one of the primary factors in decreasing their usage and developing medications that are both safe and efficient. During the research process throughout the literature, we observed that studies performed on osteoporosis using medicinal plants are still limited in the Mediterranean region. This region contains different climates, making it a suitable growth place for many medicinal plants that have proven their ability to combat many diseases. The seven medicinal plants selected in this article have proven their contributions as alternative agents in the treatment of osteoporosis and osteoporotic fractures. Throughout this review, we observed some limitations, such as a lack of clinical studies on five medicinal plants (*Ammi majus*, *Silybum marianum*, *Brassica oleracea*, *Licorice*, and *Ceratonia siliqua* L.) and molecular events that occur in in vivo studies, except for *Silybum marianum*. We recommend that future studies use biopolymers/biomaterials with these medicinal plants to promote their activity and improve osteoporotic fracture healing by improving the therapeutic targeting efficiency, reducing toxicity, and controlling sustained release. Nevertheless, the number of herbs used to treat osteoporosis is still small, and we need to increase this number by entering clinical trials. In addition to the fact that osteoporosis causes a delay in fracture healing, it also works as an enhancer of infections. Studies are still limited in this area, and we need to elaborate on the ability of these plants to function as anti-microbial agents against infections.

## Figures and Tables

**Figure 1 pharmaceuticals-17-01341-f001:**
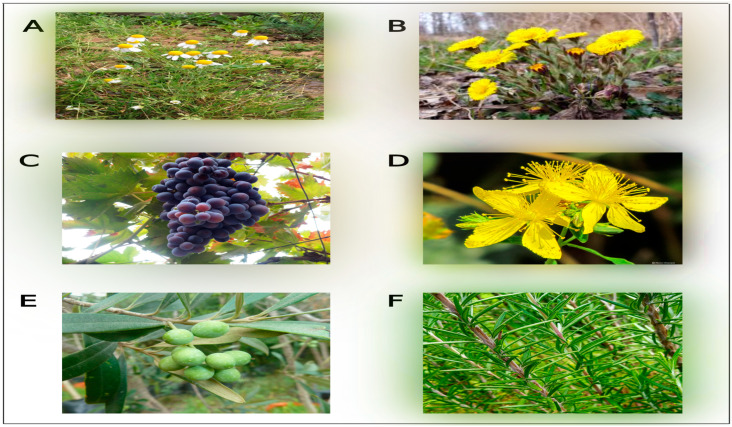
Examples of Mediterranean medicinal plants traditionally used to treat various diseases. (**A**) *Matricaria chamomilla*, (**B**) *Tussilago farfara*, (**C**) *Vitis vinifera* L., (**D**) *Hypericum perforatum*, (**E**) *Olea europaea* L., (**F**) *Rosmarinus officinalis* L.

**Figure 2 pharmaceuticals-17-01341-f002:**
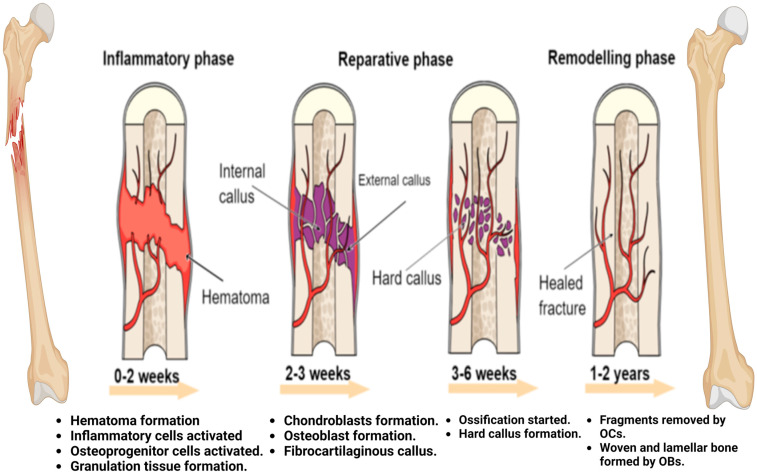
Fracture healing process, which involves three phases; (i) inflammatory phase, (ii) reparative phase, and (iii) remodeling phase. OCs: osteoclasts; OBs: osteoblasts (The illustration was generated using BioRender, www.biorender.com).

**Figure 3 pharmaceuticals-17-01341-f003:**
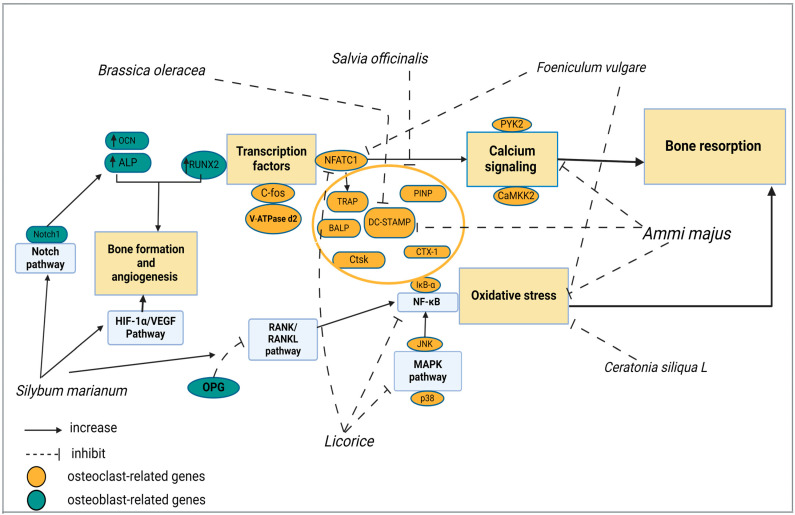
The molecular mechanisms of Mediterranean medicinal plants against osteoporosis. Bioactive components present in medicinal plants contribute to inhibiting oxidative-related signaling pathways, calcium signaling pathways, and other related genes that induce bone resorption. Also, they play a role in stimulating signaling pathways related to bone formation. TRAP: tartrate-resistant acid phosphatase; BALP: bone-specific alkaline phosphatase; CTsK: Cathepsin K; CTX-1: C-telopeptide of type I collagen; PINP: procollagen type 1 N-propeptide; DC-STAMP: dendritic cell-specific transmembrane protein; NFATc1: nuclear factor of activated T cells 1; C-fos: transcription factor; V-ATPase d2: vacuolar-type ATPase; IkB-α: inhibitor of kappa B alpha; NF-κB: nuclear factor-kappa B; JNK: Jun N-terminal kinase; p38: P 38 mitogen-activated protein kinase; MAPK: mitogen-activated protein kinase; PYK2: proline-rich tyrosine kinase 2; CaMKK2: calcium/calmodulin (CaM)-dependent protein kinase kinase 2; RANK: receptor activator of nuclear kappa-B; RANKL: receptor activator of nuclear kappa-B ligand; HIF-1α: hypoxia-inducible factor 1-alpha; VEGF: hypoxia-inducible factor 1-alpha; ALP: alkaline phosphatase; RUN-X2: Runt-related transcription factor-2.

**Table 1 pharmaceuticals-17-01341-t001:** Osteoclast-regulating factors.

Osteoclast-Regulating Factors	Functions	References
Dendritic-cell-specific transmembrane protein (DC-STAMP)/differentiation	Key regulator of the process of osteoclast formation.	[35]
Receptor activator of nuclear kappa-B (RANK)	The receptor for RANKL is responsible for regulating and activating osteoclasts.	[36]
Receptor activator of nuclear kappa-B ligand (RANKL)	Binds with the RANK receptor and plays a key part in the regulation and activation of osteoclasts.	[36]
c-fos transcriptional factor	Binds to the NFATc1 activator, leading to the activation of NFATc1, which, in response, starts the osteoclast differentiation.	[37]
Nuclear factor of activated T cells 1 (NFATc1)	Enhances osteoclast differentiation.	[37]
Tumor necrosis factor (TNF) receptor-associated factor 6 (TRAF6)	Acts as a mediator in the RANK/RANKL signaling pathway.	[38]
Macrophage colony-stimulating factor (M-CSF)	Important for differentiating osteoclasts from precursor cells to the mature form.	[39]
Osteoprotegerin (OPG)	Suppresses osteoclast formation by blocking the RANK/RANKL signaling pathway.	[36]
T cell immune regulator 1 (TCIRG1) gene	Mutations in this gene lead to the form of the a3 subunit in V-ATPase.	[40]
Vacuolar H+-ATPase (V-ATPase)	Is essential in enhancing the acidity of osteoclasts and promoting bone resorption.	[40]
Chloride channel 7 (CIC-7)	Increases osteoclast activity by promoting acidification.	[41]
Cathepsin K	Promotes the activity of osteoclasts, which stimulates organic-stage breakdown during resorption activity.	[42]

**Table 2 pharmaceuticals-17-01341-t002:** Osteoblast-regulating factors.

Osteoblast-Regulating Factors	Functions	References
Runt-related transcription factor 2 (RUNX2)	Synthesizes the proteins that play a role in the differentiation of osteoblasts.	[43]
Osterix (OSX/Sp7)	Responsible for the differentiation of osteoblasts during the embryonic stage.	[44]
Activating transcription factor 4 (ATF4)	Required for preserving bone mass by the accumulation of osteoblasts.	[45]
Collagen1-α1 (Col1a1)	Required to synthesize type 1 collagen, where type 1 collagen is secreted by osteoblasts to generate the bone matrix.	[46]
Bone γ-carboxyglutamate protein (BGLAP)	Is crucial in encoding osteocalcin.	[47]
LDL (low-density lipoprotein) receptor-related protein 5 (LRP5)	Found on the surface of osteoblasts and binds with the Wnt ligand to form a bond with osteoblasts.	[48]
SOST sclerostin (SOST)	Regulates osteoblasts by inhibiting canonical Wnt/β-catenin signaling.	[49]
β-Catenin	Controls osteoblasts by enhancing the expression of specific genes within cells (intracellular) after being activated by Wnt ligand.	[49]
Wingless-related integration site (Wnt)	The ligand binds to the surface of osteoblasts and activates β-Catenin, which regulates osteoblasts.	[49]
Core binding factor-α1 (Cbfa1)	Osteoblast differentiation.	[50]

**Table 5 pharmaceuticals-17-01341-t005:** Mediterranean medicinal plants and their anti-osteoporotic effects.

Scientific Name	Type of Extract/Bioactive Compound	Administration (In Vivo and In Vitro)	Concentrations/Doses	Effects on Bone	References
*Ammi majus*	Xanthotoxin extract.	In vitro: Bone-marrow-derived macrophages (BMM).	0.01 and 1 μM.	- Inhibited osteoclastogenesis by: Stimulating the downstream CaMKK/PYK2 signaling pathway to suppress Ca^2+^ oscillations. Suppressing ROS during osteoclast formation.	[112]
		In vivo: Ovariectomized mice model.	0.5 and 5 mg/kg (intraperitoneal injection).	- Improved histomorphometry analysis (structural and dynamic studies)	
*Ammi majus*	Methoxsalen extract	In vivo: Ovariectomized mice model.	0.02% in diet.	- Improved bone mineral density. - Upregulated osteoblasts markers (ALP, osteocalcin, and RUN-X2). - Inhibited the inflammatory reaction by decreasing related genes such as Il6 and Nfκb	[113]
*Brassica oleracea* (co-administration with *Panax ginseng*)	Hot water extract.	In vitro: RAW 264.7 cells.	50, 100, and 200 μg/mL.	- Inhibited osteoclastogenesis, especially at 200 μg/mL.	[114]
		In vivo: Ovariectomized mice model.	500 mg/kg (oral administration)	- The combination emphasized the potential of Brassica oleracea to reduce body weight and recover bone loss. - The combination decreased the number of osteoclasts.	
*Ceratonia siliqua* L.	Polyphenols	In vivo: Ovariectomized rat model.	The food was supplemented with 0.46% of the treatment.	- Development of new bone. - Increased cortical thickness and trabecular area and reduced the Haversian canal area.	[115,116]
*Ceratonia siliqua* L.	Flavonoids	In vivo: Ovariectomized rat model.	*Ceratonia siliqua* mixed with food at 3 g/kg/day/rat.	- Enhanced the bone mineral density of the proximal tibia.	[117]
*Foeniculum vulgare*	Aqueous extract of fennel	In vitro: Bone marrow-derived macrophages (BMMs).	0.5–10 μg/mL. (0.5, 1, and 2 μg/mL for the osteoclast formation test and 2 and 10 μg/mL for osteoclast activity test)	- Inhibition of osteoclast differentiation at the early stage of formation, especially at 2 μg/mL. - The activity of mature osteoclasts was impaired, especially at 10 μg/mL.	[118]
		In vivo: Ovariectomized C57BL/6 mice model.	30 and 100 mg/kg. (oral administration).	- Bone turnover decreased, reflecting microarchitecture parameters and mechanical strength improvements.	
*licorice*	Liquiritin extract.	In vitro: Bone-marrow-derived macrophages (BMMs).	0.01, 0.05, and 0.1 mM.	- The extract regulated factors promoting RANKL, such as ROS and Ca^2+^ oscillations. - Inhibition of osteoclasts by suppressing the Ca^2+^ signaling, NF-κB, and MAPK pathways.	[119]
		In vivo: Ovariectomized C57BL/6J mice models.	20 mg/kg (intraperitoneal injection).	- The extract enhanced histomorphometry parameters regarding the structural and cellular aspects.	
*licorice*	Glycyrrhiza extract (methanolic extract).	In vivo: Ovariectomized rat model.	12.4 mg/kg (oral administration).	- The extract increased bone mineral density (BMD), particularly during 6 months of administration. - The extract did not demonstrate any significant improvements in mechanical strength.	[120]
*licorice*	Formononetin extract.	In vivo: Ovariectomized rat model.	10 mg/kg	- The extract enhanced the chemical content and mechanical strength.	[121]
*Salvia officinalis*	N. D	In vivo: Ovariectomized albino rat model.	10 mg/kg (oral gavage administration).	- The anti-osteoporosis effects of this plant are reflected as follows: - Promotes osteoblastogenesis and bone formation. - Limits osteoclast function and bone resorption. - Decreases the inflammation response. - Improves histological parameters.	[122]
*Silybum marianum*	silymarin	In vivo: Ovariectomized Sprague–Dawley rat model.	100 mg/mL. (local administration).	- The extract was utilized in conjunction with hydroxyapatite (HA) because of its limited efficacy as an anti-osteoporosis agent. - The implant coated with silymarin and HA demonstrated the ability of silymarin as a good anti-osteoporotic agent. - The experiments conducted on the implant indicated successful osseointegration. - The gene expression test detected elevated levels of osteoblast gene expression, specifically OPG, Runx2, and OCN.	[123]
*Silybum marianum*	Silibinin or silybin	In vitro: Preosteoblastic cell line (MC3TE-E1).	20 and 50 μmol/L.	- The extract exhibited anti-osteoporosis efficacy by enhancing the signaling pathways associated with bone formation (1γ/VEGF and Notch) in in vitro and in vivo studies. - The analysis of the structure and histological parameters of the injured region revealed the presence of newly formed bone and tissue.	[124]
		In vivo: Ovariectomized rat model.	50 mg/kg (intraperitoneal injection).		
*Silybum marianum*	Silibinin or silybin	In vivo: Diabetic rat model.	50 and 100 mg/kg/day (oral administration).	- The anti-osteoporotic activity of the extract was demonstrated by a significant increase in all experimental parameters, especially at 100 mg/kg.	[125]

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
