# Peer review of "Evaluating the Anti-Osteoporotic Potential of Mediterranean Medicinal Plants: A Review of Current Evidence"

_pharmaceuticals, 2024, doi:10.3390/ph17101341_

Round 1
Reviewer 1 Report
Comments and Suggestions for Authors
The manuscript written by Alhareth Abdulraheem Al-A et al. shows a review about the Mediterranean medicinal plants on anti-osteoporotic. As such, the authors used various databases to search articles of Bone anatomy, osteoporotic fractures, and Mediterranean medicinal herbal therapy of osteoporotic.
However, I am confused how “the Mediterranean medicinal plants prove their ability as an alternative agent for osteoporosis and osteoporotic fractures instead of the conventional synthetic therapies”. In this manuscript, although the author tried to illustrate the curative effect of herbal medicine is superior to synthetic drugs in treating osteoporosis, there is no data to support your view. Additionally, there are no clinical data to support the therapeutic effects of herbal medicine. If such data is available, it can be incorporated into the manuscript. Moreover, it is necessary to add a figure to summarize and describe the related signaling pathways and mechanisms of effect of seven Mediterranean herbs for anti-osteoporosis. In the parts “5. Mediterranean medicinal plants with anti-osteoporosis effects”, the authors like used the description “…compared to control group...or …than the xxx group…”, could these experimental results have been done by the authors themselves, so such expressions should not be used in review writing. Authors should use appropriate language to describe the research results of cited literatures. Furthermore, the author did not cite enough literatures to review the anti-osteoporosis properties of seven Mediterranean herbs. Some herbs even have only one article supporting their anti-osteoporosis properties, which doesn't seem worth reviewing. Besides, the writing logic of the manuscript needs to be improved.
In addition, there are some points that need to be addressed by the Authors.
1. The author needs to specify which seven medicinal plants they are. (line 27, page 1; line 94, page 2)
2. In the introduction, the author has mentioned the causes of bone injury (line 40, page 1), the causes of reduced mechanical strength of bones (line 46, page 2), and the risk factors of osteoporosis (line 57, page 2). These causes are repetitious, which can easily confuse readers about the differences among them. Therefore, it is recommended that the author to rearrange this section more reasonably.
3. The content “However, the specific details of this process remain unknown” is inaccurate (line 122, page 3).
4. In Table 1, it is recommended to label the function of these factors in anti-osteoporosis and their literature sources.
5. What means “studies.” in line352, page9.
6. There is not “table 1”(on P. 14).
7. The document format of References 25 is incorrect (P. 19, Line 667).
Reviewer 2 Report
Comments and Suggestions for Authors
The submitted review “Evaluating the Anti-osteoporotic Potential of Mediterranean Medicinal Plants: A Review of Current Evidence” describes the potential of seven medicinal plants in Mediterranean regions as anti-osteoporosis agents. The manuscript has been prepared quite well.
However, the number of plants (7) with is anti-osteoporosis activity very small compared with 10.000 medicinal plant in the Mediterranean region.
The authors should provide the part of material and methods, the flow chart how did they get only 7 plants mentioned in the review. The authors should provide more medicinal plants with anti-osteoporosis effect.
Other issues:
Line 81. Aloe vera: in italic
In vivo and in vitro in italic. Check the entire manuscript
Line 310, 316, 319 : What is PG ?
Line 319-322: The sentences seems not related to topic of the review.
Line 492: Salvia officinalis in italic
Reviewer 3 Report
Comments and Suggestions for Authors
The manuscript is a review on the anti-osteoporotic potential of Mediterranean medicinal plants, discussing their use in managing osteoporosis and osteoporotic fractures.
Overall, this manuscript provides a comprehensive overview of seven Mediterranean medicinal plants that are used in osteoporosis and osteoporotic fractures by applying both in vitro and in vivo. The mechanism of action of the medicinal plants and their bioactive compounds against diseases are also briefly discussed.
The manuscript is written well and could be accepted after a minor revision mentioned below.
Comments:
The authors provided a graphic figure and several tables to establish the findings that they discussed; however, in my opinion, any in vivo experimental evidence either the authors own or published in existing literature should be provided in the manuscript.
A question is raised and needs to be answered. What factors of bioactive compounds against diseases, impact bones and impair their essential functions?
The authors mentioned that the mechanism of action of the medicinal plants and their bioactive compounds against diseases are briefly discussed. However, I couldn’t find a clear discussion regarding the mechanism of action. It could be included in a separate section to discuss.
Reviewer 4 Report
Comments and Suggestions for Authors
Major comments:-
-Improve the introduction part
-add selected plant images and mention the important phyto compounds with function ,also add one table for the plant previous studies for bone treatment
-add current and commercial treatment for bone treatment, and mention how advance of plat extract treatment for commercial and current treatment, justify and discuss
-add schematic diagram of plant extracts molecular mechanism of bone therapy
-improve the discussion part
-check all the references
-in vitro and in vivo should be italics
-add and discuss with bacterial or microbial infections in bone and its using plants
-need to expand discussion in vitro and in vivo studies and make one table
-add prisma diagram of the review
Comments on the Quality of English Languageneed improve
Reviewer 5 Report
Comments and Suggestions for Authors
In this manuscript, authors reported “Evaluating the Anti -osteoporotic Potential of Mediterranean Medicinal Plants: A Review of Current Evidence” as a review article. This manuscript is a review of the effects of the their ability as an alternative agent for osteoporosis and osteoporotic fractures instead of the conventional synthetic therapies. Although Mediterranean Medicinal Plants are familiar with people, this study provides interesting biological activities about them and some possible mechanism. These Mediterranean Medicinal Plants for osteoporotic applications contribute to the improvements of quality of life with their safety.
Given its promising but variable biological effects in this work, this compound had better to be isolated and studied by using a single compound in detail.
This manuscript should be of interest to readers interdisciplinary areas of medicinal and cosmetic researchers. Further investigations of the Mediterranean Medicinal Plants would be expected.
In summary, I think that this manuscript is appropriate for publication in Pharmaceuticals, after the following issues are addressed.
1. Abstracts: “In conclusion, the Mediterranean medicinal plants prove their ability as an alternative agent for osteoporosis and osteoporotic fractures instead of the conventional synthetic therapies. Thus, this encourages the researchers to delve deeper into this field and develop medicinal plants-based drugs.” should be corrected. “In conclusion…” should be changed more suitable one (e.g., In this review, ….).
2. Page 27, line 795, Ref 36: “Fujii, T., Murata, K., Mun, S. H., Bae, S., Lee, Y. J., Pannellini, T., ... & Ivashkiv, L. B. “MEF2C regulates osteoclastogenesis and pathologic bone resorption via c-FOS,” Bone Res. 2021, 9, no. 1. doi: 10.1038/s41413-020-00120-2. “Authors should confirm all references. The “….” may be incorrect. (Pannellini, T., ... & Ivashkiv, L. B.)
3. Introduction: Some important review on the Medicinal Mediterranean Plants is missing. (e.g., Journal of Ethnopharmacogy, 2008, 116, 341-357; Complementary and Alternative Therapies and the Aging Population, 2009, 541-562…), there will exists a variety of plants in the Mediterranean Area, includes rosemary, licorice, chamomile and olive oil among others. Authors should add some pictures of these plants to introduction section or TOC.
Round 2
Reviewer 1 Report
Comments and Suggestions for Authors
The revised manuscript can be accepted, since it has been revsed according to the comments.
Author Response
The revised manuscript can be accepted, since it has been revsed according to the comments.
We would like to thank you for your effort in helping me to improve the quality of this paper. Thank you for your comments
Reviewer 2 Report
Comments and Suggestions for Authors
The authors have modified their manuscript, added the methodology section, Current and commercial therapies in osteoporosis as well as a table of medicinal plants from different regions of the world with their anti osteoporotic properties.
However, the authors did not showedin details how they searched and got only 7 medicinal plants in the Mediterranean region. It seems that, the authors have only selected quite popular medicinal plants to review, so the manuscript did not give much information about the medicinal plants with anti osteoporotic properties in the Mediterranean region. With Google scholar, I easily found olive oil for prevention of bone loss.
I recommend the manuscript should be rejected.
Reviewer 4 Report
Comments and Suggestions for Authors
Accept
Comments on the Quality of English Languageneed minor edit
